# Unpredictable benefits of social information can lead to the evolution of individual differences in social learning

Pieter van den Berg [1,2] ✉, TuongVan Vu[3] & Lucas Molleman [4]

Human ecological success is often attributed to our capacity for social learning, which facilitates the spread of adaptive behaviours through populations. All humans rely on social learning to acquire culture, but there is substantial variation across societies, between individuals and over developmental time. However, it is unclear why these differences exist. Here, we present an evolutionary model showing that individual variation in social learning can emerge if the benefits of social learning are unpredictable. Unpredictability selects for flexible developmental programmes that allow individuals to update their reliance on social learning based on previous experiences. This developmental flexibility, in turn, causes some individuals in a population to end up consistently relying more heavily on social learning than others. We demonstrate this core evolutionary mechanism across three scenarios of increasing complexity, investigating the impact of different sources of uncertainty about the usefulness of social learning. Our results show how evolution can shape how individuals learn to learn from others, with potentially profound effects on cultural diversity.

Social learning is a cornerstone of human adaptability. It empowers us to adapt quickly to new circumstances while avoiding the costs of trial and error, and fosters the dissemination of valuable cultural knowledge through populations[1–9]. Yet, reliance on social learning is neither indiscriminate nor universal. Individuals often use social information strategically, most notably by preferentially learning from others who perform well[10–13]. The degree to which we rely on social learning varies widely, between societies[14–19], between individuals within societies[20–26], and across development[18,27–30]. Theoretical models from across the biological and social sciences show that this variation is important for collective dynamics: individual differences in social learning affect consensus-building, cooperation, and coordination of actions in groups, modulating the speed of diffusion of innovations in social networks, and impacting the direction and outcome of cultural evolution[31–42]. However, it is still unclear why variation in reliance on social learning exists[15,23,24].

We address this question with an evolutionary model that challenges the conventional model assumption that individuals have hardcoded and immutable social learning strategies[1,2] (for reviews, see[43,44]). Instead, we focus on the evolution of factors that shape reliance on social learning throughout development. Our model allows for the evolution of flexible developmental programs, in which individuals can adjust their reliance on social learning depending on their past learning experiences. This approach formalises recent ideas that social learning itself may be learned[15,45,46]. Individual differences in reliance on social learning may result from such flexible developmental programs, depending on the extent to which individuals experience different circumstances.

Generally speaking, we expect that developmental flexibility in social learning will evolve if social learning is beneficial to some individuals but not to all. Under such conditions, flexibility allows individuals to learn from experience whether a strong reliance on social

[1]KU Leuven, Department of Biology, Leuven, Belgium. [2]KU Leuven, Department of Microbial and Molecular Systems, Leuven, Belgium. [3]Department of Clinical, Neuro-, & Developmental Psychology, Vrije Universiteit Amsterdam, Amsterdam, The Netherlands. [4]Department of Psychology, University of Amsterdam, Amsterdam, The Netherlands. ✉e-mail: piet.vandenberg@kuleuven.be

learning is beneficial to them. To investigate this general idea, we consider several different scenarios. We start out with a basic scenario in which social learning is simply more effective for some individuals than for others. Next, we move on to scenarios where all individuals learn equally effectively, but may get different payoffs from adopting the same cultural traits (e.g., beliefs, norms, or material artefacts). This relaxes the common yet unrealistic premise of classic cultural evolution models that all individuals receive the same expected payoffs from any particular cultural trait[1,2,43,47]. We expand on this scenario in a final step in which we investigate the impact of assortment, where individuals are more likely to learn from others who receive similar payoffs from adopting cultural traits.

Our results show a consistent pattern: flexible developmental programs evolve if the value of social information is uncertain. This flexibility then leads to individual differences in reliance on social learning in the population, with some individuals ending up heavily relying on social learning and others relying on individual learning. In contrast, we do not observe the evolution of flexibility when uncertainty about the value of social information is relatively low, leading to populations that consist of only rigid social learners or only rigid individual learners.

## Results
### The model
We constructed an individual-based simulation model, explicitly simulating the evolution of a population of 1000 individuals over 10,000 generations to study how individual differences in social learning might emerge over evolutionary time. Each individual in the model must repeatedly decide whether to adopt cultural traits. They do this by assessing whether the trait is beneficial (i.e., has a positive payoff), using either individual or social learning. The degree to which individuals rely on social (rather than individual) learning is not fixed, but is allowed to depend on experience. For example, if an individual adopts a beneficial trait after social learning, they may become more reliant on social learning in the future. In our model, the degree to which individuals make such adjustments to their reliance on social learning is determined by genes that are under selection throughout the generations. Table 1 gives an overview of all model parameters.

In our model, the degree to which an individual relies on social learning (denoted $S$) can change over the individual's lifetime. 'Reliance on social learning' is defined as the probability that an individual learns socially (rather than individually) when they must decide whether to adopt a cultural trait. It is determined by two genes: the individual's initial reliance on social learning ($S_O$) and their developmental flexibility in reliance on social learning ($\Delta$). $S_O$ determines the probability that the individual uses social learning the first time that they consider adopting a cultural trait. With the complementary probability $1 - S_O$, they will rely on individual learning. 'Developmental flexibility' (or just 'flexibility'; $\Delta$) indicates how much an individual changes their reliance on social learning based on learning experiences (see Fig. 1). The higher $\Delta$, the more strongly the individual takes past experience into account in updating $S$. If $\Delta$ is high, the individual develops a stronger reliance on whichever type of learning has proven effective in previous learning experiences. Note that while we call $S_O$ and $\Delta$ 'genes', we do not mean to suggest that these two aspects are encoded as single genes in the human genome. Rather, this term reflects a common simplifying assumption about the genetic causation of these traits that mutations create continuous variation and that both traits mutate independently[48].

At the start of the evolutionary simulations, all individuals have a value of $\Delta$ that is equal to 0, which means they are inflexible - their reliance on social learning remains equal to $S_O$ throughout their lifetimes. From one generation to the next, both $S_O$ and $\Delta$ can change through mutations, producing new developmental programs that might fare better (or worse) than existing ones. There is a small fitness cost associated with $\Delta$ to ensure that high levels of $\Delta$ only evolve if there are clear fitness benefits from being flexible (see Methods for details and Table 1 for an overview of all model parameters; simulations with an absent or higher cost show qualitatively similar results - see Supplementary Fig S1). In sum, the genes $S_O$ and $\Delta$ do not directly determine the individual's reliance on social learning but rather specify a developmental programme that shapes reliance on social learning in response to experience over time.

Over their lifetimes, individuals must sequentially decide whether to adopt ten different cultural traits, each time relying on either individual or social learning. If they learn individually, they form an imperfect ('noisy') impression of the payoff that they would achieve from adopting the trait, e.g., by experimentation (Fig. 1a). If they learn socially, they make an - again imperfect - assessment of the payoff that ten random other individuals obtain for adopting that trait. Socially learning from multiple individuals in principle allows individuals to increase the accuracy of their payoff assessments. However, it depends on the effectiveness of social learning (which can in turn depend on the constitution of the population, see Results) whether social learning will indeed be more effective than individual learning for any given individual (Supplementary Note 1 presents a comparison of assessment accuracies of individual learning and social learning in various settings). For both forms of learning, individuals adopt a trait only if they assess it to be beneficial (i.e., if its associated payoff is positive; see Methods for further details). Half of the cultural traits confer a benefit while the other half are detrimental, and all cultural traits are beneficial or detrimental to the same degree (i.e., all traits have payoffs of either 1 or −1).

After the ten learning events, all individuals reproduce proportionally to the total payoff accrued from adopting cultural traits. Their offspring inherit their values of $S_O$ and $\Delta$ with a small chance of mutation (causing random changes with a small amount; see Methods). After reproduction, the same cycle starts again with the offspring generation. After 10,000 generations, we observe how $S_O$ and $\Delta$ have evolved, and assess the presence of individual differences in reliance on social learning at the end of development in the final generation. For each parameter combination that we consider, we report summary statistics of 200 independent simulations runs (replicates).

### Impact of the effectiveness of social learning
We start with a basic scenario in which we assume that there may be differences between individuals in the effectiveness of social learning. We use the term 'effectiveness' to indicate how well social learning helps an individual determine whether adopting a cultural trait would be beneficial or detrimental to them. When effectiveness is high, social learning is a useful way to ascertain whether cultural traits will be beneficial, while a low effectiveness indicates that social learning will often fail to accurately determine the payoff consequences associated with adopting cultural traits. Individual learning is equally effective across all simulations (see Methods for details), so a lower effectiveness of social learning indicates a higher relative effectiveness of individual learning. There are various reasons why there may be individual differences in the effectiveness of social learning: they may reflect differences in social learning ability, or local environmental differences that make social information easier to interpret for some individuals than for others. We assume that the differences in learning ability are not heritable, but our results likely hold qualitatively for low values of heritability.

This scenario serves as a first investigation into whether individual differences in reliance on social learning can evolve. The reasoning in this case is simple. If social learning is more effective for some individuals than for others, then flexibility may be advantageous, because it can allow more effective social learners to become more reliant on social learning during development, while simultaneously allowing less effective social learners to reduce their reliance on social learning. In

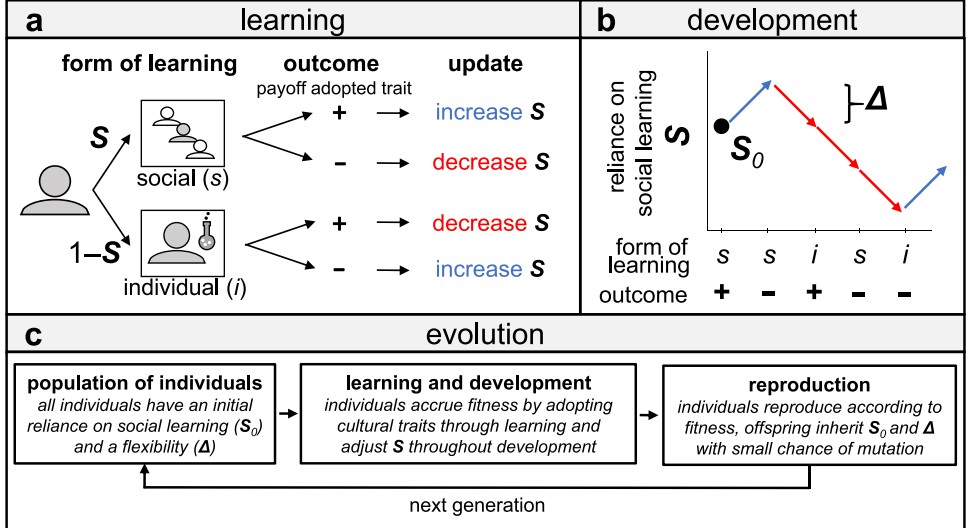

**Fig. 1 | Model overview. a** During their lifetimes, individuals repeatedly consider adopting cultural traits based on either social learning (with probability $S$) or individual learning (with probability $1-S$). Their tendency to rely on social learning is initially equal to $S_O$, but can change based on outcomes of learning, depending on their flexibility (determined by $\Delta$). After a positive experience with social learning, individuals increase their reliance on social learning by magnitude $\Delta$. After a negative experience with social learning, it decreases by $\Delta$. The opposite holds for individual learning: $S$ decreases with $\Delta$ after a positive experience with individual learning, and increases with $\Delta$ after a negative one. As a result, individuals increase their reliance on the type of learning that they just had a positive experience with. **b** Example trajectory of an individual's reliance on social learning ($S$) throughout development. In our evolutionary model, both individuals' initial reliance on social learning ($S_O$) and magnitude of flexibility ($\Delta$) are subject to natural selection (Methods). **c** Evolutionary framework in which developmental variables $S_O$ and $\Delta$ may change over time through natural selection.

sum, if there are individual differences in the effectiveness of social learning, flexibility may evolve.

In our simulations, we systematically vary both the average effectiveness of social learning in the population, and the degree to which this effectiveness differs between individuals in the population. The results show that a high average effectiveness of social learning leads to a high reliance on social learning and low flexibility (purple lines in Fig. 2a, b, see Fig. 2c for evolutionary trajectories of $S_O$ and $\Delta$). Low average effectiveness of social learning leads to the evolution of low reliance on social learning, and, again, low flexibility (blue lines in Fig. 2a, b, evolutionary trajectories in Fig. 2i). These results are not very surprising: if social learning generally works well, it evolves, and if it does not, it does not. Flexibility does not evolve in these cases, which means that the individuals in these simulations do not change their reliance on social learning much over their development (see developmental trajectories in Fig. 2d, j). Because of this, the populations in these simulations do not end up harbouring individual differences in reliance on social learning (see developmental outcomes in Fig. 2e, k).

When the average effectiveness of social learning is intermediate, the picture is different (yellow lines in Fig. 2a, b). Now, the evolution of reliance on social learning ($S_O$) and flexibility ($\Delta$) depend on whether there are individual differences in the effectiveness of social learning. If there is a high degree of individual variation (x-axis in Fig. 2a, b), this means that social learning is highly effective for some individuals but not at all effective for some others. If this is the case, we see that developmental flexibility in reliance on social learning evolves (Fig. 2b), because it allows individuals with higher effectiveness to increase their reliance on social learning throughout development (as they are likely to adopt beneficial traits and avoid adopting detrimental traits), while it allows individuals with low effectiveness to reduce their reliance on social learning and become more reliant on individual learning instead (Fig. 2g). As a result of this evolved flexibility in social learning, we observe clear individual differences in reliance on social learning by the end of development (Fig. 2h).

This first set of simulations shows that individual differences in reliance on social learning can evolve when individuals face uncertainty on whether it will be more beneficial for them to use individual or social learning. If this uncertainty is large enough, evolution will lead to flexible developmental programs allowing individuals to adjust their reliance on social learning based on experience. This flexibility in turn leads to the emergence of individual differences in reliance on social learning. By contrast, inflexible learning strategies evolve in environments where the benefits of social learning are predictably high or low, leading to homogenous populations of individuals who all heavily rely on either social or individual learning, respectively. An alternative implementation of our model in which individuals do not adjust their reliance on social learning directly but rather update their expectations of the payoffs of both types of learning leads to the same conclusions (Supplementary Fig S3).

### Impact of payoff differences

In a next step, we investigate whether flexibility in reliance on social learning can evolve if the differences in the effectiveness of social learning are not directly assumed, but rather arise as a natural consequence of the constitution of the population. In this scenario, we no longer assume that there are individual differences in the effectiveness of social learning in assessing the payoffs of cultural traits. Instead, we now assume that individuals may obtain different payoffs from adopting cultural traits. It is still the case that half of the cultural traits are detrimental and half are beneficial for all individuals, but which specific traits are detrimental or beneficial can now vary between individuals (i.e., they can have different 'payoff profiles'). This means that the same cultural traits may now be beneficial to some individuals but detrimental to others, which can cause social learning to lead to the adoption of detrimental traits. There may be several reasons why individuals would receive different payoffs from cultural traits: individuals might face different local conditions (e.g., so that they differ in their goals, needs or preferences[43,49–53]), they may have experienced different life events (e.g., early-life adversity enhancing behaviours with short-term benefits[54]), or their existing repertoire of cultural traits may differ, causing differences in compatibility of new cultural traits (e.g., a chosen profession or position in society rendering the

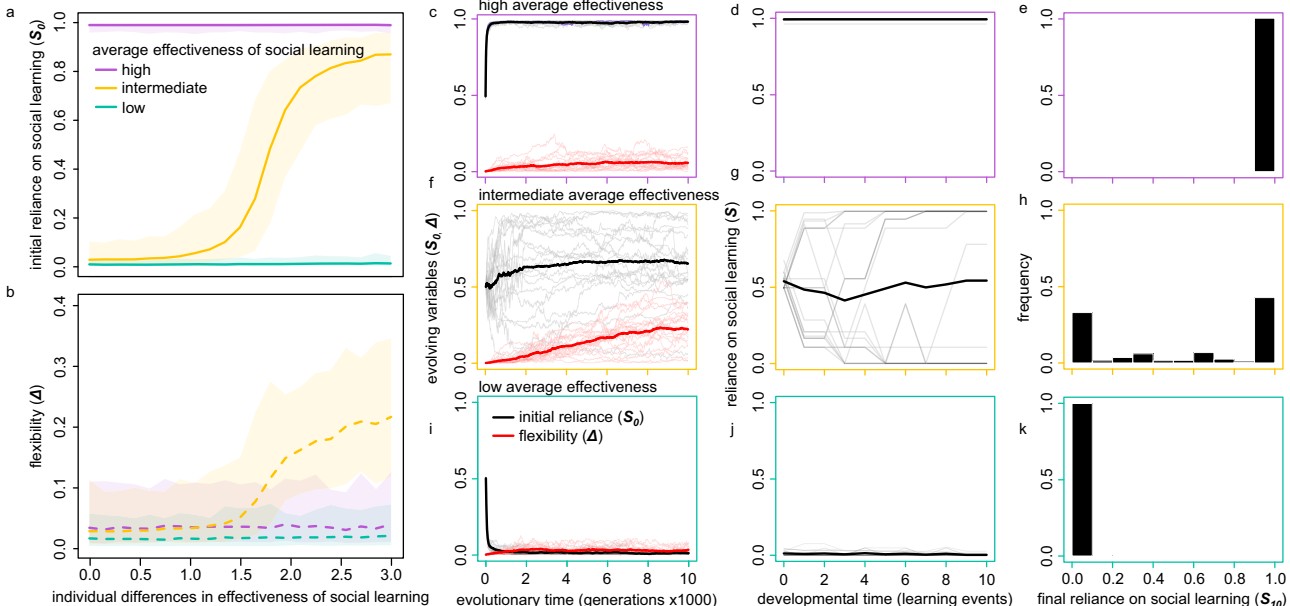

**Fig. 2 | Flexibility and individual differences in reliance on social learning evolve when social learning pays for some but not for others. a** Evolved initial reliance on social learning ($S_O$) and (**b**), flexibility in reliance on social learning ($\Delta$), depending on the average (three lines) and variation (x-axis) in the effectiveness of social learning. Different mean effectiveness levels are indicated by different coloured lines. The degree of individual variation in effectiveness is on the horizontal axis. Lines indicate average outcomes over 200 replicate simulation runs for each parameter combination; shading indicates the standard deviation. **c**, **f**, **i** Evolutionary trajectories over 10,000 generations of the population averages of initial reliance on social learning ($S_O$, black lines) and flexibility in reliance on social learning ($\Delta$, red lines) for high (**c**), intermediate (**f**), and low (**i**) average social learning effectiveness. Graphs show evolutionary trajectories from 20 randomly chosen simulation runs (thin lines) for each evolving variable, and their averages

(thick lines). **d**, **g**, **j** Individual developmental trajectories of reliance on social learning over 10 learning events, for high (**d**), intermediate (**g**), and low (**j**) social learning effectiveness. Graphs show trajectories of 20 individuals who were randomly chosen from the population at the end of a single representative simulation, as well as the averages over these subsets of individuals (thick lines). **e**, **h**, **k** Histograms of the reliance on social learning at the end of development ($S_{10}$). Bars show binned fractions of reliance on social learning across the entire population of the same simulation as the corresponding graphs (**d**, **g**, **j**). For graphs (**c**–**k**), the individual variation in social learning effectiveness is equal to 2.0 (see Methods for details). See Supplementary Fig. S2 for full distributions of simulation outcomes for low, intermediate and high average effectiveness of social learning, across the full range of individual differences in effectiveness of social learning.

expression of certain traits socially inappropriate[55,56]). The individual payoff profiles are assumed not to be heritable (see Methods for details).

To investigate under which circumstances flexibility and individual differences in reliance on social learning may evolve, we compare populations that differ in the degree to which payoff profiles vary between individuals. In this context, the degree of 'individual variation' reflects the probability that any two individuals receive a different payoff for any given cultural trait. In addition, we vary the degree of 'clustering' of these individual differences. Clustering refers to the distribution pattern of these individual differences within the population, indicating the extent to which there is grouping with respect to the payoffs individuals receive (see Methods for details). In a population that has highly clustered individual differences, there are essentially two types of individuals who have exactly opposite payoff profiles (e.g., populations with a high degree of specialisation or distinct ideological or cultural divides, causing payoffs to be similar within groups but different between them). When clustering is low, the differences are less pronounced—any two individuals will likely have payoff profiles that partially overlap (e.g., populations with less social structure or professional populations with high degrees of interdisciplinarity).

Figure 3 provides an overview of how differences in payoff profiles and clustering of these differences affect evolutionary outcomes. When there is little variation in the payoff profiles, evolution leads to high levels of initial reliance on social learning and low flexibility (Fig. 3a, left-hand side). This makes sense: when all individuals in the population receive the same payoffs from cultural traits, social learning allows individuals to adopt beneficial traits and avoid detrimental

traits. Conversely, when individual variation is high, social learning is less effective, so we observe the evolution of low initial reliance on social learning and low flexibility (Fig. 3a, right-hand side). These results indicate that heavy reliance on social learning might evolve in some populations but not in others, depending on the degree to which people receive different payoffs from adopting cultural traits.

Developmental flexibility in social learning ($\Delta$) can evolve in populations where the variation is in an intermediate range around 0.1 (Fig. 3a, darker shading). This is especially the case when payoff differences are highly clustered (Fig. 3b), while flexibility does not evolve when clustering is low (Fig. 3c). We can see why this happens when we consider a population with a variation of 0.1 and maximal clustering. This population effectively consists of a majority of 90% of the individuals who share the same payoff profile, and a minority of the other 10% for whom the profile is exactly opposite. Social learning is effective for the individuals who belong to the majority, because they are likely to learn from others who receive similar payoffs as they do. In contrast, social learning is ineffective for minority individuals, because they are likely to learn from majority individuals who receive the exact opposite payoffs. As a consequence, these minority individuals are better off learning individually. These conditions favour the evolution of flexibility, because it allows the individuals who belong to the majority to increase their reliance on social learning over development, while simultaneously allowing individuals that belong to the minority to reduce their reliance on social learning. Individual differences in reliance on social learning are the ultimate outcome of this (see Supplementary Fig S4). When clustering is low, there is no majority-minority situation - all individuals tend to resemble all other individuals to roughly the same degree. This renders social learning approximately

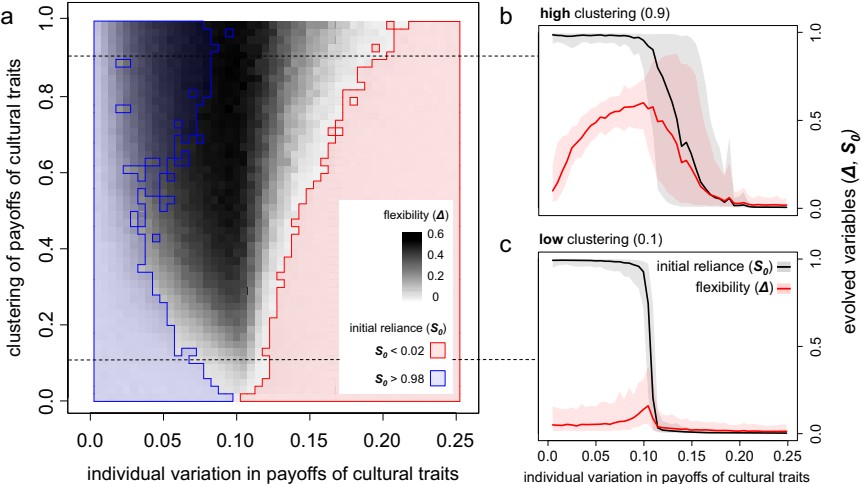

**Fig. 3 | The evolution of flexibility in reliance on social learning when individuals obtain different payoffs from adopting cultural traits. a** Evolved flexibility in reliance on social learning in different populations. Populations differ in the degree of individual variation in payoff consequences of cultural traits, and the clustering of these differences. The blue- and red-shaded areas indicate the populations for which initial reliance on social learning evolves to high and low levels, respectively. In the remaining area, initial reliance on social learning evolves to intermediate levels, and relatively high levels of flexibility often evolve (darker shading refers to stronger flexibility). **b** Evolution of initial reliance on social learning and flexibility when clustering is high (0.9) and (**c**), low (0.1). The lines indicate means over 200 replicate simulations and the shading indicates the standard deviation. See Supplementary Fig. S5 for full distributions of simulation outcomes for high and low clustering.

equally effective (or ineffective) for all individuals, so flexibility in social learning is not favoured by natural selection.

### Impact of assortment in social learning

In a final step, we build further on the previous scenario where individuals have different payoff profiles, but now allow for assortment in social learning. 'Assortment' here refers to the degree to which individuals preferentially learn from others with similar payoff profiles. For example, this may be the case in 'echo chambers' where individuals that hold similar beliefs interact[57]—these individuals may receive more similar payoffs from new beliefs because of compatibility with their existing belief system. If there is little assortment, we are close to the previous scenario where individuals learn from others more or less at random (e.g., leftmost inset in Fig. 4). If assortment is intermediate (i.e., equal to 1), individuals are more likely to learn from more similar others, with a probability that is proportional to this similarity. For example, they are twice as likely to learn from individuals that are twice as similar (middle inset in Fig. 4). If assortment is high, individuals practically always learn from those that they share the highest similarity with (e.g., rightmost inset in Fig. 4). We focus on the situation where the variation in payoffs that individuals obtain from cultural traits is equal to 0.2 and the clustering is 0.5 (cf. Fig. 3).

Figure 4 shows the impact of assortment on the evolution of $S_O$ and $\Delta$. We observe that intermediate values of assortment lead to the highest flexibility. The explanation for this follows a similar logic as before. When individuals hardly assort, they often end up learning from others that have quite different payoff profiles. As a result, social learning does not benefit anyone in the population, and low reliance on social learning evolves (with little flexibility). Conversely, with high assortment, individuals learn from others who have similar payoff profiles, and hence social learning tends to pay off well for all individuals in the population. This leads to the evolution of a strong reliance on social learning, again with low flexibility. When assortment is intermediate, social learning will work well for individuals that are generally more similar to others in the population, but it is not very beneficial for those who are relatively dissimilar to most others. Despite intermediate assortment, the latter group will still select dissimilar individuals as models too often for social learning to be effective. Under these circumstances, we observe the evolution of

intermediate (to high) initial reliance on social learning, and high flexibility. As before, this ultimately results in the emergence of individual differences in social learning in the population (see Supplementary Fig S6).

Across all scenarios we investigated, we see a consistent pattern. If circumstances render social learning beneficial for all individuals in the population, a strong reliance on social learning will evolve, and flexibility does not provide any advantage. Because flexibility is costly, such circumstances lead to the evolution of a homogeneous population of inflexible individuals. A similar pattern holds if circumstances imply low effectiveness of social learning across the entire population: reliance on social learning evolves to low levels, and the resulting population is homogeneous and inflexible. In circumstances where some individuals fare well when they rely on social learning, but others are better off learning individually, flexibility has a selective advantage. If this is the case, flexibility allows individuals to learn from experience whether social learning works well for them, and to adjust their reliance on social learning accordingly. This flexibility will lead some individuals to become heavily reliant on social learning while others become individual learners. In other words, under such circumstances evolution leads to the emergence of individual differences in reliance on social learning.

## Discussion

Our model provides a general explanation for the widely observed but hitherto poorly understood phenomenon of individual differences in social learning. Our simulation results show how natural selection can shape individuals' capacities to flexibly adjust their reliance on social learning based on their experiences. This developmental flexibility can evolve when there is variation in the effectiveness of social learning (Fig. 2), or when the payoffs associated with cultural traits vary between individuals, either in the absence of assortment (Fig. 3) or in its presence (Fig. 4). When individuals go through different experiences with individual and social learning, developmental flexibility can lead some individuals in a population to develop a heavy reliance on social learning, while others end up consistently resorting to individual learning.

Our model formalises recent ideas that social learning strategies may themselves be learned[15,45,46]. We assumed that individuals' genes

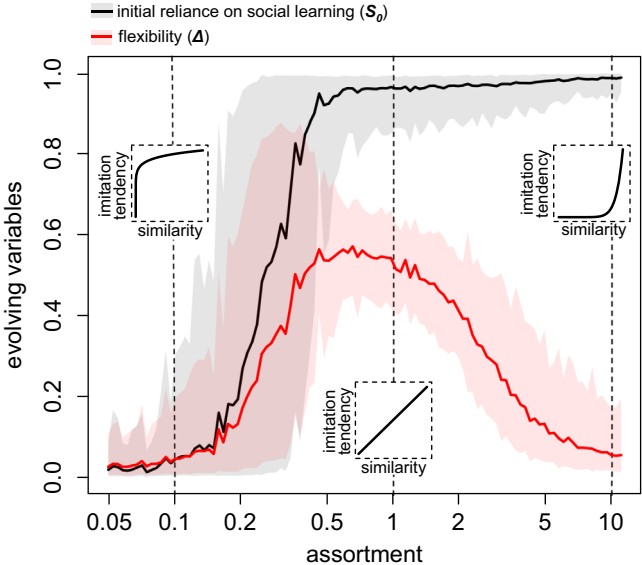

**Fig. 4 | The evolution of initial reliance on social learning and flexibility depending on the degree of assortment.** Evolved values at the end of the evolutionary simulation. The graph shows the mean (solid line) and standard deviation (shading) over 200 replicate simulation runs. These are results for a population where the individual variation in payoffs received from cultural traits is equal to 0.2 and the clustering is equal to 0.5 (cf. Fig. 3). Insets show characterizations of assortment for the values 0.1, 1 and 10 (from left to right), indicating the relative likelihood that an individual will learn from another individual, depending on the similarity in payoffs that they receive from adopting cultural traits (with the x- and y-axis both ranging from 0 to 1).

do not directly encode fixed learning strategies (as traditionally assumed[1,2]), but determine a developmental programme enabling individuals to adjust their reliance on social or individual learning based on experience. In our model, fixed learning strategies do evolve, but only in environments where the value of social information was predictably low or high (Figs. 2–4). In such cases, individuals do not benefit from flexibility, because the "information environment" is stable enough to be tracked by processes that occur at longer time scales (like genetic evolution; see[58–63] for further discussion on the impact of uncertainty on adaptation). This result may help explain widely observed between-population differences in reliance on social learning[15,19,64]: our model predicts reliance on social learning to be high in societies where payoffs for behaviours are homogeneous (e.g., when variation in local environmental conditions is low, or due to "cultural tightness" discouraging deviant behaviour[65]), and lower in societies where they are more heterogeneous (e.g., if local environments vary, if there are more (sub)cultural differences, or when individuals have disparate needs or preferences[50–53]).

Environments in which the value of social information is unpredictable favour the evolution of developmental flexibility. In these cases, at the beginning of their lives, individuals cannot be sure whether or not social learning will be beneficial to them, and flexibility ($\Delta$) helps them adjust their learning strategy to their personal circumstances. Put more generally, the evolution of developmental flexibility in our model is an example of plasticity being favoured by natural selection in the face of uncertainty[61]. Our model shows that for a trait like social learning, this uncertainty can be rooted in the constitution of the population (the social environment), leading to the evolution of flexibility without the need for uncertainty about the external environment, with individual differences in reliance on social learning as a result. For example, our model predicts that individuals who belong to a majority group should rely more strongly on social learning than those who belong

to minority groups (Fig. 3b), as long as assortment is not too strong (in which case our model predicts both groups to strongly rely on social learning; see Fig. 4). We suspect that a tendency to learn from those who obtain similar payoffs might be beneficial in mixed populations (potentially reducing the evolutionary scope for flexibility)−extending our model to study this could be an interesting avenue for further research.

Our modelling approach allowed us to use established methods from genetic evolution to model cross-generational transmission of the variables defining developmental programmes ($S_O$ and $\Delta$). However, these traits too might be subject to other forms of transmission, including teaching and forms of social learning[15,45,46,66]. Our model can serve as a template for exploring these alternative possibilities of the transmission of the variables that determine flexibility in reliance on social learning, their consequences for the evolution of learning strategies, the speed of populations adapting to environmental change, and the direction and outcome of cultural evolution[15,38,42].

In our model, the individual differences in reliance on social learning that arise tend to be extreme, with some individuals ending up always relying on social learning and others becoming exclusive individual learners. This is partly a result of our modelling setup, where we favoured several simplifying assumptions over more realistic assumptions. Individuals always increase (or decrease) reliance on social learning by $\Delta$, regardless of their current reliance on social learning, and the effectiveness of social learning and individual learning does not differ depending on the cultural trait in question. In reality, social learning might be more effective in some situations while individual learning might be better in others. Relaxing these assumptions would provide an interesting avenue for further research, and might provide insight into how more gradual variation in reliance on social learning might evolve.

Individual differences in reliance on social learning arise in our model based on experiences with learning. The individual differences resulting from this process might be further enhanced by other mechanisms, like specialisation[67]: individual learners might become good innovators, and social learners might expand their social networks and acquire even more useful social information. Another possibility is that individuals' effectiveness of individual and social learning strengthen each other (e.g., because these forms of learning share the same underlying mechanisms[11,68–70]). Future work may examine these scenarios by extending our model, mapping out the scope for developmental flexibility and individual variation in social learning when learning effectiveness depends on individuals' experience level.

## Methods
### Overall setup
Individuals in the first generation of each simulation were constructed by drawing a number from a uniform distribution between 0 and 1 to initialise $S_O$ and initialising $\Delta$ at 0. We ran all simulations for 10,000 non-overlapping generations, and reported the outcomes (evolved $S_O$ and $\Delta$) in the final generation. All simulation codes were written in C++. See Table 1 for an overview of all model parameters.

In each generation, individuals go through ten sequential learning events. For each learning event, they first decide whether to use social learning (with probability $S$) or individual learning (with probability $1-S$), and then determine whether to adopt the cultural traits using the chosen type of learning (see below). Individuals receive payoffs from adopted cultural traits (either −1 or 1), while they receive no payoff from not adopting a cultural trait. Individuals also pay a small fitness penalty for flexibility: 0.01 multiplied by their value of $\Delta$ is subtracted from their total accumulated payoff. After the ten learning events, individuals reproduce proportionally to the payoffs they have accumulated, passing their values for $S_O$ and $\Delta$ on to their offspring with a small chance of mutation.

**Table 1 | Overview of model parameters**

| parameter | values |
|---|---|
| Population size | 1000 individuals |
| Simulation length | 10,000 generations |
| Number of learning events | 10 events |
| Number of social learning models | 10 models |
| Initial $\Delta$ | 0 |
| Initial $S_O$ | Uniformly drawn from [0,1] per individual |
| Cost of flexibility | 0.01 |
| Mutation probability $\Delta$ | 0.001 |
| Mutation probability $S_O$ | 0.001 |
| Mutation step size $\Delta$ | 0.05 |
| Mutation step size $S_O$ | 0.05 |
| Standard deviation of noise on individual learning | 1.0 |
| Standard deviation of noise on social learning | Scenario 1: varied<br>Scenario 2: 1.0<br>Scenario 3: 1.0 |
| Individual variation in payoffs of cultural traits | Scenario 1: N/A<br>Scenario 2: varied<br>Scenario 3: 0.2 |
| Clustering of payoffs of cultural traits | Scenario 1: N/A<br>Scenario 2: varied<br>Scenario 3: 0.5 |
| Assortment | Scenario 1: N/A<br>Scenario 2: N/A<br>Scenario 3: varied |

Mutations at both loci occur independently of each other and with a probability that is equal to 0.001 in both cases. If a mutation occurs, a random number drawn from a normal distribution with mean 0 and standard deviation 0.05 is added to the parental value to determine the offspring value (we obtain qualitatively similar patterns when mutations are taken from a uniform distribution between 0 and 1; see Supplementary Fig S7). If $S_O$ would mutate outside of the possible range between 0 and 1, it is set to 0 or 1 instead. If $\Delta$ would mutate below 0, it is instead set to 0. Mutations introduce new variation in $S_O$ and $\Delta$ into the population, which allows for incremental evolutionary change over the generations. To be clear, the final reliance on social learning by the end of development ($S_{10}$) has no bearing on inheritance - only $S_O$ and $\Delta$ are inherited.

**Adopting cultural traits based on social or individual learning**
Over their lifetimes, individuals consecutively consider whether to adopt a total of ten cultural traits. For each individual, five of the ten traits confer a fitness benefit (payoff of +1), whereas the other five are detrimental to fitness (payoff of −1; we obtain qualitatively similar results for more continuously variable payoffs, see Supplementary Fig. S7). Individuals receive a baseline payoff of 5 to ensure that their fitness cannot be negative after all ten learning events. Individuals decide whether to adopt a cultural trait based on either individual or social learning. For both forms of learning, individuals can make imperfect assessments of the payoffs that would be obtained if the trait were adopted. Specifically, individual learning consists of drawing such an assessed payoff from a normal distribution centred on the actual payoff and with standard deviation 1 (the fact that payoffs are always equal to −1 or 1 implies a probability of 0.159 of incorrectly concluding that a detrimental trait is beneficial or vice versa). Social learning consists of making such assessments for ten random other individuals and then averaging these numbers (in the case of assortment, these individuals are drawn non-randomly, see below). For social learning, the standard deviation of the payoff assessment is also equal to 1 (except in

the scenario where we varied social learning effectiveness, see below). See Supplementary Note 1 for an analysis of the relative effectiveness of social and individual learning across our different scenarios.

**Updating reliance on social learning**
When considering whether to adopt the first of the ten cultural traits, the probability $S$ that an individual will rely on social learning is given by $S_O$, which can take any value between 0 and 1. After any learning experience, an individual updates its value of $S$ with magnitude $\Delta$ (cf. Fig. 1). Specifically, an individual adds $\Delta$ to their current value of $S$ after a positive social learning experience or a negative individual learning experience, and subtracts $\Delta$ after a negative social learning experience or a positive individual learning experience. If adding or subtracting $\Delta$ from $S$ would lead to a value under 0 or exceeding 1, it is set to 0 or 1 instead. See Supplementary Methods and Supplementary Fig. S3 for an alternative implementation of updating that leads to qualitatively similar conclusions.

**Scenario 1: effectiveness of social learning**
In the first scenario, we varied the effectiveness of social learning by varying the standard deviation of the noise on the payoff assessment during social learning (see above). When this noise has a larger standard deviation, social learning is less effective because observed payoffs do not accurately reflect the actual payoffs. In our simulations, we set this standard deviation to 0.1 for high effectiveness, 3.5 for intermediate effectiveness, and 5.0 for low effectiveness. To allow for individual differences in the effectiveness of social learning, we drew a number from a gamma distribution for each individual to determine their individual level of noise in payoff assessments. This gamma distribution had its mean centred on the average social learning effectiveness in the population (0.1, 3.5 or 5.0; see above), while its standard deviation determined the degree to which there are individual differences in social learning effectiveness (cf. x-axis in Fig. 2a, b).

**Scenario 2: payoff differences**
In the second scenario, individuals receive different payoffs from adopting cultural traits. We investigated both the effect of the amount of individual variation in payoff profiles and the effect of clustering of these profiles on the evolution of flexibility in reliance on social learning (see Fig. 3). To implement this, we drew a number from a beta distribution for each individual to determine the payoffs that they receive from adopting the cultural traits (the 'payoff profile'). This number can vary between the extremes of 0 (where the individual receives positive payoffs for all odd traits—traits 1, 3, 5, 7, and 9, and negative payoffs for all even traits—traits 0, 2, 4, 6, and 8) and 1 (where the payoff profile is exactly opposite). Intermediate values lead to an intermediate payoff profile, where the individual receives positive payoffs for some odd traits and some even traits, and negative payoffs for the others. To be exact, the number drawn determines the probability for each of the five positive payoffs to occur among the even traits. For example, if the value is 0.5, the individual likely has about two or three out of the five positive payoffs among the even traits, and the rest among the odd traits. If this beta distribution has a mean close to 0, most individuals draw numbers close to 0, and therefore have similar payoff profiles. As we increase the mean towards 0.25 (cf Fig. 3), the individual variation in the population increases. Individual variation would again decrease after 0.5, but we only considered the range between 0 and 0.25 since individual learning is already clearly dominant at 0.25. To increase clustering, we increased the variance of the beta distribution. An intermediate mean and a high variance then leads to most individuals drawing numbers close to either 0 or 1, which results in a 'polarised' population where there are essentially two groups who receive opposite payoffs. If the mean is intermediate and variance low, most individuals draw a number close to the mean, causing the individual differences to be more distributed. The variance

is bound between a minimum of 0 and a maximum of mean * (1−mean) −in Fig. 3a, the y-axis indicates the fraction of the maximal variance.

## Scenario 3: assortment

In the final scenario, we built further on scenario 2 by allowing assortment in social learning. Specifically, we assumed that the individual variation in payoffs received from cultural traits was equal to 0.2 and the clustering was equal to 0.5 (see above). In addition, individuals no longer learn from random other individuals in the population, but they are now more likely to learn from those that have similar payoff profiles. This similarity is simply given by the proportion of cultural traits that the individuals receive the same payoff for. In this scenario, individuals first randomly draw 100 potential individuals to learn from, and then select ten individuals out of these 100 based on similarity and the degree of assortment. Specifically, the similarities are all taken to the power of the assortment, and then ten individuals are selected proportionally to the resulting values. These are the ten others that the individual will potentially socially learn from for all ten cultural traits. Hence, if assortment is 0, all similarities are taken to the power 0, which yields 1 regardless of similarity, causing the individual to learn from a random subset of individuals. If assortment is equal to 1, the probability to learn from another individual is proportional to similarity. If assortment is very high, then the similarities closer to 1 will get a disproportionately heavy weight, causing individuals to learn almost exclusively from others that are most similar to them.

## Simulation setup

We ran 200 replicate simulation runs for each parameter combination: for the different means and standard deviations in social learning effectiveness in Fig. 2, for the different degrees of variation in payoffs and clustering in Fig. 3, and for the different degrees of assortment in Fig. 4.

## Reporting summary

Further information on research design is available in the Nature Portfolio Reporting Summary linked to this article.

## Data availability

The simulation data generated in this study have been deposited in a repository of the Open Science Framework, https://doi.org/10.17605/OSF.IO/7TA9M.

## Code availability

The code for producing the simulation data of this study has been deposited in a repository of the Open Science Framework, https://doi.org/10.17605/OSF.IO/7TA9M.

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

## Acknowledgements

This work was supported by the Research Foundation—Flanders (FWO) [Senior Postdoctoral Mandate 12W3821N to PVDB]. This work benefited from extensive comments by Wouter van den Bos, Jörg Gross, Björn Lindström, Johannes Ullrich, and the members of the Connected Minds lab at the University of Amsterdam. The resources and services used in this work were provided by the VSC (Flemish Supercomputer Centre), funded by the Research Foundation—Flanders (FWO) and the Flemish Government.

## Author contributions

PvdB: Conceptualization, model implementation and analysis, visualization, writing lead. TVV: Conceptualization, reviewing and editing. LM: Conceptualization, analysis, co-writing, reviewing and editing.

## Competing interests

The authors declare no competing interests.
