## [Peer Review File · Nature Communications]

Individual differences in social learning can evolve when the benefits of social information are unpredictableReviewers' Comments:

Reviewer #1:

Remarks to the Author:

This paper reports on a new agent-based model of the evolution of social learning reliance and its flexibility. The main results illustrate that when social information is universally valuable or low value, flexibility in social learning will not evolve. On the other hand, in situations where the efficacy of social learning is variable across individuals, it behooves agents to be flexible in their reliance on social learning – learning within their lifetimes how valuable it is to rely on social learning. The authors elaborate this model to illustrate how certain social realities like individual differences in which cultural traits are beneficial, the extent to which these payoffs are clustered (which they term polarization), and assortment along these payoff similarities, will change the value of plasticity in social-learning reliance. Again, the intuitions they arrive at are similar – when this results in individual variation in the efficacy of social learning (for example, as is the case in a highly polarized environment with intermediate individual variation in payoffs), flexibility will be favored by natural selection.

This is one of the clearest agent-based models I've read recently, and the authors do a commendable job of explaining the logic behind each of their results. This is unfortunately all too rare in simulation articles. The authors also clearly justify the model in light of some empirical observations regarding within-population and between-population variation in the reliance on social learning.

I recommend the paper be published, but have two suggestions for how the paper can be improved to increase its impact.

- 1) The links to the empirical literature should be spelled out more clearly. For example, there sometimes seems to be a mismatch in the framing that purports to explain cross-cultural variation in reliance on social learning, while the analysis of the model results overwhelmingly focuses on explaining individual variation in social learning reliance within populations. I suggest the discussion flesh out some predictions, or perhaps make sense of some existing empirical literature, on the types of contexts and individuals who are likely to rely on social learning.
 - a. For example, at the group level: lower individual variation in the payoff of cultural traits should lead to more social learning in more homogenous societies. Would you predict that in societies with less specialization there should be more social learning then? Could this account for the seemingly irrationally low levels of social learning participants engage in, in highly specialized societies? Could there be other operationalizations of homogeneity in payoffs? Would this help make sense of some of Mesoudi et. al.'s results of greater social learning reliance in China compared to Western countries?
 - b. For example, at the individual level: do we have evidence that minorities are less likely to socially learn than majorities in polarized societies? In less polarized contexts do these distinctions go away?
- 2) On the more purely theoretical front, I couldn't help but think that many of these results were not particularly specific to the trait in question being social learning. Rather the framework and the intuitions driving the results seem to be more generally about the contexts in which plasticity is favored. Could you spell out more clearly what is it about this model that makes it specific to social learning? Or put another way, can you contrast the results of this model to one where the trait whose plasticity is evolving is not plasticity?

Minor points

- Line 273 – replace “that” with “who”
- Line 436 – why “may be”?
- Line 336-341 – This seems like an important observation about the consequences of the way the delta variable is implemented. Could you make the learning process a bit more dynamic wherein individuals are more Bayesian in their updating within their lifetimes? Would this change the dynamics in any meaningful way? – e.g., Frankenhuys, W. E., & Panchanathan, K. (2011). Balancing sampling and specialization: an adaptationist model of incremental development. *Proceedings of the Royal Society B: Biological Sciences*, 278(1724), 3558-3565.

- Similarly, do you think it would matter to your conclusions if the agents made adjustments proportional to how positive or negative an experience they had had with social and individual learning?

Cristina Moya

Reviewer #2:

Remarks to the Author:

The paper presents a model on cultural learning that aims to explain under what circumstances interindividual differences in learning strategies (individual learning vs. social learning) can emerge. The authors propose that people can differ in the return from adopting a certain 'practice' and that such differences can change the viability or value of social learning. In the extreme cases, the model shows when and why people may only learn from own experiences, only through social learning, or some people rely on social learning whereas others rely on individual learning.

I found the model quite interesting. By relaxing an often "given" assumption of cultural learning (namely, that adopting "what others do" is always beneficial for oneself), the model can reveal when cultural learning should occur, how cultural learning may even lead to assortment and group segregation, and why individual learning can still be advantageous in certain situations.

In general, I also found the model elegant and simple from a first glance, in the sense that with few stylized assumptions, it can generate some interesting observations. At the same time, and after a deeper consideration, there were quite a lot of parameters in the model that made it a bit complicated to judge how general the observed patterns were or to which degree the fixed parameters would influence the dynamics.

Overall, I think the model makes an interesting contribution that can inspire also empirical follow-up work on the trade-off between individual and cultural learning and could provide a general framework on the integration of individual and social learning and maybe even the emergence of groups and group boundaries (related to trends, different 'cultures' within a society, maybe even diverging norms etc.) based on misaligned benefits for certain actions. I do, however, have some more technical questions about the general results outlined in more detail, below.

(1) Penalty parameter.

The authors write that they introduced a small fitness cost associated with delta. This parameter seemed to have been fixed to a value throughout all simulations. I was therefore wondering what would happen if this parameter is set to 0 or a higher value. In the former case, is it always optimal (i.e., a dominant strategy) to adopt maximum flexibility?

(2) Social learning efficiency.

Another parameter that seems to be fixed without much discussion is the number of agents that one learns from (fixed to ten random other individuals). In general, it was not so clear to me where the advantage of social learning comes from (when payoffs are aligned); see also point below. It would be more transparent, if actual payoff calculations would be presented to show how much social learning can, for example, reduce noise in learning and increase expected payoffs (compared to individual learning) and how this advantage may turn into a disadvantage when payoffs become less aligned (and systematic social learning mistakes increase).

(3) Benchmarks of "optimal" strategies.

Having a benchmark of what an "optimal learner" should do could also give some insights into how good agents are in "exploiting" their environment. What I mean by that is the following: In the

optimal case, agents would earn 10 "fitness points" per iteration/generation, if I understood correctly. If learning errors are low and everybody has aligned "preferences", agents should be able to get close to this maximum through social learning (it is unclear to me, how much they would earn in expectation through individual learning, though – yet, this should be easy to calculate?). With misaligned preferences, agents seem to face a trade-off between more noisy individual learning and potentially misleading (i.e., wrong) social learning. This trade-off could be made much more transparent, I think, with theoretical benchmarks for some strategies, like how much a pure social learner vs. individual learner would earn in different environments. Again, from my reading, with introducing payoff misalignment, agents now need to balance the efficiency of social learning (i.e., reduced noise, which maybe could be quantified better) and the probability of making mistakes (i.e., adopting a trait that is detrimental) and this balance (if I am not mistaken here) could be quantified and presented a bit better.

More generally, how much evolving agents earn would be interesting to show across the different scenarios. This would also allow to see how close they get to the theoretically possible maximum earnings and how much misalignment leads to potentially sub-optimal earnings.

(4) Going beyond ABM.

Relatedly, I was also wondering whether some baseline model could be solved analytically. When simplifying some assumptions (e.g., the imperfect assessment, see below), it should be relatively easy (but I hope I am not mistaken here), to calculate the expected payoffs of pure individual learners and pure social learners and compare that as two distinct strategies and see under which condition what strategy outperforms the other. As far as I understand, this is not a game-theoretic model in the sense that payoffs depend on the strategies that other agents use in the population. Rather, payoffs depend on some (fixed) parameters in the environment – which would mean that expected payoff calculations should already tell what strategy should dominate the other (i.e., regardless of the composition in the population).

Essentially, the model proposes that the value from "social learning" can vary, based on the correlation between payoffs across agents. It would be nice to see at what point the expected payoffs from social learning start to become smaller than the expected payoffs from individual learning, given the alignment or misalignment in payoffs in the population. This should be possible to simply calculate, I think, and would give straightforward "cutoff" points at which, for example, social learning is not viable anymore (or for what agent-type is not worthwhile).

(5) Imperfect assessment.

At some points, I was wondering whether the implementation of parameters is a bit overcomplicated. For example, the authors chose an operationalization of "imperfect assessments" based on a normal distribution with a standard deviation of 1. Here, I was wondering why the authors not simply introduce a probability of making a learning mistake. For example, there could be probability p that an agent learns that a trait will generate a payoff of +1, whereas it actually generates a payoff of -1. I guess by drawing from a normal distribution, this can be reformulated into a probability, but it would be easier to interpret for the reader if this could be translated into an error probability. This could also make expected payoff calculations much easier (again, I hope I am not missing something crucial here).

(6) It would be nice in the methods or SI to have an overview of all parameters in the model and, for fixed parameters, to which values they were fixed (e.g., a simple table).

(7) In the ABM, individuals adopt new strategies based on accumulated payoffs and random mutation. Random mutations are important to see if, in a population of homogenous agents, different strategies can 'invade' and take over.

However, in the implementation, random mutation is based on the parental values and a small deviation, based on a normal distribution. This makes it more likely that random mutants are "close" to what is already in the population (in terms of their parameters). Often, random mutations are just taken from a uniform distribution over the entire possible strategy space to see if also "distant"

strategies can outperform the existing agents and invade. I was wondering if that is a problem for the simulations in this case. In the long run, it may not matter much but in the short run, the simulations may end up in a local maximum that is hard to get out of, because mutants have a small likelihood to adapt a vastly different strategy than the residents. Given the results, I do not think it is a big problem, but maybe this could be double checked (or explained why the authors chose this implementation for their mutants).

(8) It was not clear to me whether payoff-contingencies (i.e., for which trait an agent received a positive/negative payoff) were inherited or maybe even completely fixed throughout generations. This could be maybe more clearly communicated.

(minor)

Page 14: "we increased the variance of this ..." (small typo)

Page 3: The authors often refer to the two parameters as "genes" here. From my understanding, the authors do not literally believe that the S and delta parameter are literally encoded as genes in humans. Maybe it would make sense to more clearly communicate on this page that "genes" is more like a metaphor here for two independent strategy parameters and should not be taken to literally.

Reviewer #3:

Remarks to the Author:

Comments to Author

This paper is almost impossible to read. The terms used in the paper: "efficacy of social learning", "flexibility", "polarization", "assortment", "reliance on social learning", total of 200 replicates (line 451), 20 randomly chosen applicates (line 171), 1,000 individuals (line 67), 20 randomly chosen individuals (line 173).

Figure 2, which is very hard to read, purports to show much of the evolution of S, which appears to be the probability of learning socially. But the additional term "efficiency of social learning" is not defined in the text, but seems to turn up in Methods (line 399). Here, however, it has a separate paragraph which does not mention what "average social learning efficacy" is nor what "individual variation" in it is. It is "graphed" in Figure 2a, b, but again there is no definition. Does it relate to payoffs? Scenario 1 in Methods is totally confusing.

Scenario 2 in Methods introduces polarization without a qualitative justification. Under what empirical situation would this be relevant? On the other hand, assortment seems to be related to what in most of the literature is called "conformity bias" or "frequency dependent bias". Does this lead to polarization?

The confusion over social learning efficiency is amplified in lines 187-198: "We assume all individuals have an equal social learning efficiency". But on line 199-206 it is differences in "reliance on social learning" that matter. Also, how is "individual variation" in social reliance assessed to start with? Again, on line 228, what is "developmental flexibility"? Is it just Δ ?

In the inheritance of social learning (S and Δ are referred to as genes, which they obviously are not), it appears that generations are non-overlapping, but what are the properties of S and Δ at the outset of generation $t+1$ in terms of the parental generation t ? Is there a new value of S_0 at the beginning of each generation? Does it depend on variation in S_{-10} from generation t ? How is Δ inherited if at all?

The authors do not seem to be aware of Wakano et al. Theor. Popul. Biol. 66: 249-258 or Aoki et al. Curr. Anthropol. 46: 334-340 on the evolution of social learning in varying environments.

The title of the paper is misleading. The assumptions, although very confusing, seem to include many about individual differences, for example, lines 366-371 and 379-389. With these assumptions that there are individual differences, how can they "emerge"?

Abstract, lines 19-20. The two papers referred to above answered the question by showing that if environments are very predictable or very unpredictable then social learning has no advantage, while in-between it does have an advantage.

The kind of simulation study where you throw in the whole "kitchen sink" of parameters and variables is very unlikely to yield useful qualitative conclusions. This paper does not advance theory of social evolution.

Response to reviewers for the manuscript ‘Individual differences in social learning emerge through the evolution of developmental flexibility’

By Pieter van den Berg, TuongVan Vu, and Lucas Molleman

Below, we reprint all comments and suggestions of the reviewers in blue. Our responses, printed in black, include a full guide to the changes we have made to our manuscript in response to their concerns.

Reviewer #1

This paper reports on a new agent-based model of the evolution of social learning reliance and its flexibility. The main results illustrate that when social information is universally valuable or low value, flexibility in social learning will not evolve. On the other hand, in situations where the efficacy of social learning is variable across individuals, it behooves agents to be flexible in their reliance on social learning – learning within their lifetimes how valuable it is to rely on social learning. The authors elaborate this model to illustrate how certain social realities like individual differences in which cultural traits are beneficial, the extent to which these payoffs are clustered (which they term polarization), and assortment along these payoff similarities, will change the value of plasticity in social-learning reliance. Again, the intuitions they arrive at are similar – when this results in individual variation in the efficacy of social learning (for example, as is the case in a highly polarized environment with intermediate individual variation in payoffs), flexibility will be favored by natural selection.

This is one of the clearest agent-based models I’ve read recently, and the authors do a commendable job of explaining the logic behind each of their results. This is unfortunately all too rare in simulation articles. The authors also clearly justify the model in light of some empirical observations regarding within-population and between-population variation in the reliance on social learning.

We thank Prof. Moya for her appreciation of our study and her kind words. In response to her comments below, we have made several changes. We have more clearly positioned our results in the existing literature, especially regarding between-population differences and the evolution of plasticity in general. We have also developed a new version of our model with a more mechanistically conceived implementation of developmental flexibility in reliance on social learning, and have used this model to investigate whether our results hold when payoffs of cultural traits vary continuously (they do). Details are under the individual comments below.

We thought it was a good suggestion to change ‘polarization’ to ‘clustering’ in the manuscript, so we have updated this throughout (also given the comments of Reviewer 3 that they found this term opaque).

I recommend the paper be published, but have two suggestions for how the paper can be improved to increase its impact.

1) The links to the empirical literature should be spelled out more clearly. For example, there sometimes seems to be a mismatch in the framing that purports to explain cross-cultural

variation in reliance on social learning, while the analysis of the model results overwhelmingly focuses on explaining individual variation in social learning reliance within populations. I suggest the discussion flesh out some predictions, or perhaps make sense of some existing empirical literature, on the types of contexts and individuals who are likely to rely on social learning.

a. For example, at the group level: lower individual variation in the payoff of cultural traits should lead to more social learning in more homogenous societies. Would you predict that in societies with less specialization there should be more social learning then? Could this account for the seemingly irrationally low levels of social learning participants engage in, in highly specialized societies? Could there be other operationalizations of homogeneity in payoffs? Would this help make sense of some of Mesoudi et. al.'s results of greater social learning reliance in China compared to Western countries?

We thank the reviewer for these thoughtful questions. In our initial submission we put the evolution of individual differences in reliance on social learning centre stage, as we felt that this was the most novel aspect of our study. We agree with the reviewer that our paper benefits from highlighting more how our model results speak to the widely documented between-population differences in reliance on social learning. In our revision, we have now emphasised our results on between-population differences throughout, and made links with existing literature where appropriate. In light of this comment, we added a sentence to the Introduction that highlights between-group differences (lines 62-65), and also emphasised these results more in the Results section for each scenario (lines 212-215; 259-261; 308-313). In addition, we reflect on the population-level results in our revised Discussion (lines 364-370). There, we discuss how payoff correlations might help explain cross-societal variation in social learning, and speculate what aspects of cultures might correlate with reliance on social learning, such as cultural “tightness”. We feel that the presentation in the revision is more balanced.

b. For example, at the individual level: do we have evidence that minorities are less likely to socially learn than majorities in polarized societies? In less polarized contexts do these distinctions go away?

To the best of our knowledge, there is no empirical research that shows that minorities learn differently than majorities. On this topic, we only know of Mesoudi 2018 (“Migration, acculturation, and the maintenance of between-group cultural variation” Plos One; <https://doi.org/10.1371/journal.pone.0205573>), which is a more theoretical paper that is also pointing out a lack of empirical work on this topic. In our revised Discussion section on this point (lines 376-383), we now reflect on the predictions of our model with regard to individual differences in more detail. It is true that our model from scenario 2 predicts that majority individuals should rely more on social learning than minority individuals, but scenario 3 shows that we would only expect this if there is not too much assortment within the majority and minority groups. We now outline these predictions in the Discussion section (lines 376-383).

2) On the more purely theoretical front, I couldn't help but think that many of these results were not particularly specific to the trait in question being social learning. Rather the framework and the intuitions driving the results seem to be more generally about the

contexts in which plasticity is favored. Could you spell out more clearly what is it about this model that makes it specific to social learning? Or put another way, can you contrast the results of this model to one where the trait whose plasticity is evolving is not plasticity?

This is an interesting perspective, and we agree that some of our results pertain to the evolution of plasticity more broadly. Specifically, this is the case for the results of our first model, presented in figure 2 (where there are intrinsic differences between individuals in their social learning effectiveness). In this first model, we could simply interpret individual and social learning as two different phenotypes (1 and 2), and the ‘social learning effectiveness’ as the payoff of phenotype 1 compared to that of phenotype 2. We would then have flexibility (=plasticity) evolving if individuals have sufficient uncertainty about which of the phenotypes would confer them the best payoff.

However, in the later models that we present in figures 3 and 4, it is specifically relevant that our model considers flexibility in social learning, not just plasticity in general. The reason for this is that the fitness associated with social learning is dependent on what others in the population are doing. Since we specifically consider differences in population constitution in these models, we here obtain insights that are specific for flexibility in social learning.

We have rewritten the relevant paragraph in the Discussion to consider to what extent our model reflects more general findings with respect to plasticity, and where it provides more specific insights for flexibility in social learning in particular (lines 371-383).

Minor points

- Line 273 – replace “that” with “who”

This change has been made.

- Line 436 – why “may be”?

This has been changed to ‘are now’.

- Line 336-341 – This seems like an important observation about the consequences of the way the delta variable is implemented. Could you make the learning process a bit more dynamic wherein individuals are more Bayesian in their updating within their lifetimes? Would this change the dynamics in any meaningful way? – e.g., Frankenhuis, W. E., & Panchanathan, K. (2011). Balancing sampling and specialization: an adaptationist model of incremental development. *Proceedings of the Royal Society B: Biological Sciences*, 278(1724), 3558-3565.

It is indeed true that our model implements flexibility in the simplest possible way. Following the reviewer’s suggestion, we have implemented an alternative model in which the individuals are essentially Bayesian reinforcement learners. However, after careful consideration, we have deviated from the suggestion to implement the approach in Frankenhuis & Panchanathan (2011), who compute optimal developmental programs. That paper considers a much simpler scenario where the individual develops towards either of two phenotypes, or samples a cue. In our case, the situation is more complicated: the “phenotypic specialisation” (development towards either social or individual learning) affects

the information that the individual obtains in the future (in Frankenhuis & Panchanathan, there is no such feedback of phenotype on information gathering). Hence, we made the judgement that it would be beyond the scope of this paper to implement this.

We instead implemented a version of the model in which all individuals have separate beliefs about the expected payoffs associated with social learning and individual learning. After each learning event, the individual updates the expected payoff associated with the type of learning they just employed (governed by Δ). The reliance on social learning is now determined by the difference between the expected payoffs of social and individual learning, and a softmax function determines how strongly this difference impacts the probability that an individual chooses either form of learning.

We ran this model across a range of parameter conditions for the first scenario of our paper. The results are in close agreement with our original model. A more detailed description of this model and its results can be found in the Supplementary Materials (Supplementary Methods and Figure S2). We also briefly discuss this model in the Results section of the main text (lines 215-218).

- Similarly, do you think it would matter to your conclusions if the agents made adjustments proportional to how positive or negative an experience they had had with social and individual learning?

The model outlined under the previous point of the reviewer actually allows us to answer this question, as the individuals are now keeping track of the expected payoffs of both types of learning. We used this model to rerun our simulations for scenario 1 (differences in the effectiveness of social learning), but now the payoffs are normally distributed around 1 and -1 (rather than always being equal to 1 or -1), with standard deviation of 0.5. We obtain qualitatively similar results to those reported in our main analyses: flexibility evolves for intermediate average effectiveness of social learning and high individual variation in effectiveness. We now present these results in the Supplementary Materials (Supplementary Figure S6) and also refer to them in the Methods section (lines 440-442).

Cristina Moya

Reviewer #2

The paper presents a model on cultural learning that aims to explain under what circumstances interindividual differences in learning strategies (individual learning vs. social learning) can emerge. The authors propose that people can differ in the return from adopting a certain 'practice' and that such differences can change the viability or value of social learning. In the extreme cases, the model shows when and why people may only learn from own experiences, only through social learning, or some people rely on social learning whereas others rely on individual learning.

I found the model quite interesting. By relaxing an often "given" assumption of cultural learning (namely, that adopting "what others do" is always beneficial for oneself), the model can reveal when cultural learning should occur, how cultural learning may even lead to

assortment and group segregation, and why individual learning can still be advantageous in certain situations.

In general, I also found the model elegant and simple from a first glance, in the sense that with few stylized assumptions, it can generate some interesting observations. At the same time, and after a deeper consideration, there were quite a lot of parameters in the model that made it a bit complicated to judge how general the observed patterns were or to which degree the fixed parameters would influence the dynamics.

Overall, I think the model makes an interesting contribution that can inspire also empirical follow-up work on the trade-off between individual and cultural learning and could provide a general framework on the integration of individual and social learning and maybe even the emergence of groups and group boundaries (related to trends, different 'cultures' within a society, maybe even diverging norms etc.) based on misaligned benefits for certain actions. I do, however, have some more technical questions about the general results outlined in more detail, below.

We thank the reviewer for their appreciation for the paper and their constructive comments. We are specifically thankful for the comments below, which have provided a number of important checks of the robustness of our results with respect to specific assumptions. We have run a range of additional simulations to check the robustness of our results with respect to the parameters and implementations mentioned in the reviewer's comments, and generally find that our results are highly robust. In addition, we present an analytic version of our model in the Supplementary Materials, providing benchmarks for our simulations and helping the reader build intuitions about the trade-offs in our model. We feel that this has substantially strengthened our paper. We go into more detail under the specific points below.

(1) Penalty parameter.

The authors write that they introduced a small fitness cost associated with delta. This parameter seemed to have been fixed to a value throughout all simulations. I was therefore wondering what would happen if this parameter is set to 0 or a higher value. In the former case, is it always optimal (i.e., a dominant strategy) to adopt maximum flexibility?

We thank the reviewer for this valuable suggestion. In response, we have rerun the simulations for Scenario 1 in the main text, now considering both the complete absence of a cost of flexibility, and double the cost of flexibility. In both cases, we obtain qualitatively similar results as in Figure 2 of the main text: flexibility only evolves for intermediate effectiveness of social learning, and if individual variation in social learning effectiveness is sufficiently high. Hence, it is not the case that maximum flexibility is always the most advantageous in the absence of a cost. We now present these results in Supplementary Figure S1 and refer to the fact that the results are robust with respect to this cost in The Model section (lines 98-101).

(2) Social learning efficiency.

Another parameter that seems to be fixed without much discussion is the number of agents that one learns from (fixed to ten random other individuals). In general, it was not so clear to me where the advantage of social learning comes from (when payoffs are aligned); see also point below. It would be more transparent, if actual payoff calculations would be presented to

show how much social learning can, for example, reduce noise in learning and increase expected payoffs (compared to individual learning) and how this advantage may turn into a disadvantage when payoffs become less aligned (and systematic social learning mistakes increase).

When introducing our model, we now make explicit where the advantage of social learning might come from (*i.e.*, from reducing noise; lines 124-129). In addition, in response to this point and several of the points of this reviewer below, we also devised an extensive additional model analysis with the purpose of specifically showing how the relative performances of individual and social learning compare to each other across the different parameter combinations of our three scenarios (see Supplementary Note). This also includes a description of how the accuracy of social learning depends on sample size. We believe that this addition helps the reader build an intuition about what is going on in our model.

(3) Benchmarks of “optimal” strategies.

Having a benchmark of what an “optimal learner” should do could also give some insights into how good agents are in “exploiting” their environment. What I mean by that is the following: In the optimal case, agents would earn 10 “fitness points” per iteration/generation, if I understood correctly. If learning errors are low and everybody has aligned “preferences”, agents should be able to get close to this maximum through social learning (it is unclear to me, how much they would earn in expectation through individual learning, though – yet, this should be easy to calculate?). With misaligned preferences, agents seem to face a trade-off between more noisy individual learning and potentially misleading (*i.e.*, wrong) social learning. This trade-off could be made much more transparent, I think, with theoretical benchmarks for some strategies, like how much a pure social learner vs. individual learner would earn in different environments. Again, from my reading, with introducing payoff misalignment, agents now need to balance the efficiency of social learning (*i.e.*, reduced noise, which maybe could be quantified better) and the probability of making mistakes (*i.e.*, adopting a trait that is detrimental) and this balance (if I am not mistaken here) could be quantified and presented a bit better.

More generally, how much evolving agents earn would be interesting to show across the different scenarios. This would also allow to see how close they get to the theoretically possible maximum earnings and how much misalignment leads to potentially sub-optimal earnings.

We fully agree that this was missing from our paper and needed to be addressed - we thank the reviewer for pointing this out. In the model analysis in the Supplementary Note we provide extensive benchmarks, characterising performance of social learning across each of our three scenarios, and compare that with individual learning. We believe that adding these benchmarks will help the reader gain intuitions about the trade-offs that operate in our model (*e.g.*, involving reducing noise and payoff misalignment).

(4) Going beyond ABM.

Relatedly, I was also wondering whether some baseline model could be solved analytically. When simplifying some assumptions (*e.g.*, the imperfect assessment, see below), it should be relatively easy (but I hope I am not mistaken here), to calculate the expected payoffs of

pure individual learners and pure social learners and compare that as two distinct strategies and see under which condition what strategy outperforms the other. As far as I understand, this is not a game-theoretic model in the sense that payoffs depend on the strategies that other agents use in the population. Rather, payoffs depend on some (fixed) parameters in the environment – which would mean that expected payoff calculations should already tell what strategy should dominate the other (i.e., regardless of the composition in the population).

Essentially, the model proposes that the value from “social learning” can vary, based on the correlation between payoffs across agents. It would be nice to see at what point the expected payoffs from social learning start to become smaller than the expected payoffs from individual learning, given the alignment or misalignment in payoffs in the population. This should be possible to simply calculate, I think, and would give straightforward “cutoff” points at which, for example, social learning is not viable anymore (or for what agent-type is not worthwhile).

The model analysis in the Supplementary Note mentioned above provides an extensive comparison of expected payoffs of individual learning and social learning across the three scenarios we consider. We believe that this analysis will help the reader build intuition about our model.

(5) Imperfect assessment.

At some points, I was wondering whether the implementation of parameters is a bit overcomplicated. For example, the authors chose an operationalization of “imperfect assessments” based on a normal distribution with a standard deviation of 1. Here, I was wondering why the authors not simply introduce a probability of making a learning mistake. For example, there could be probability p that an agent learns that a trait will generate a payoff of +1, whereas it actually generates a payoff of -1. I guess by drawing from a normal distribution, this can be reformulated into a probability, but it would be easier to interpret for the reader if this could be translated into an error probability. This could also make expected payoff calculations much easier (again, I hope I am not missing something crucial here).

It is true that drawing values from a normal distribution essentially equates to an error probability in the case of a single learning event, as is the case with individual learning in our model. Because an individual learner draws from a normal distribution with the mean centered on the true value of the cultural trait (1 or -1), and a standard deviation of 1, individual learners always have a probability of about 0.16 of making an error. For social learning however, this doesn't quite hold in such a simple way. The multiple social observations are averaged and this then leads to a probability of making a mistake. As a result, the uncertainty associated with a single observation and the probability of making a mistake are not very intuitively related, so framing everything in terms of probability to make a mistake is not necessarily simpler or easier to interpret. Because of this, we considered it most straightforward to stick with the current implementation, but we now have pointed out that individual learning leads to mistakes in about 16% of the cases in the Methods section (lines 446-450). Additionally, as mentioned above, the new Supplementary Note now outlines in detail how the relative performances of both types of learning depend on the payoffs that the individuals obtain for adopting cultural traits.

(6) It would be nice in the methods or SI to have an overview of all parameters in the model and, for fixed parameters, to which values they were fixed (e.g., a simple table).

We thank the reviewer for this suggestion - it will make our paper more easily readable. We have now added a Table with all model parameters at the end of the Methods section.

(7) In the ABM, individuals adopt new strategies based on accumulated payoffs and random mutation. Random mutations are important to see if, in a population of homogenous agents, different strategies can 'invade' and take over.

However, in the implementation, random mutation is based on the parental values and a small deviation, based on a normal distribution. This makes it more likely that random mutants are "close" to what is already in the population (in terms of their parameters). Often, random mutations are just taken from a uniform distribution over the entire possible strategy space to see if also "distant" strategies can outperform the existing agents and invade. I was wondering if that is a problem for the simulations in this case. In the long run, it may not matter much but in the short run, the simulations may end up in a local maximum that is hard to get out of, because mutants have a small likelihood to adapt a vastly different strategy than the residents. Given the results, I do not think it is a big problem, but maybe this could be double checked (or explained why the authors chose this implementation for their mutants).

This is a fair point - it is indeed possible for simulations to end up on local peaks because they are constrained in how they mutate. We reran our simulations for scenario 1 with an alternative implementation of mutation, where any mutation of either flexibility or initial reliance on social learning results in a value that is drawn uniformly from the interval between 0 and 1. The results of these simulations are in accordance with the conclusions presented in the main text and can be found in Supplementary Fig S5. We now also refer to the results of these simulations in the Methods section (lines 431-432).

(8) It was not clear to me whether payoff-contingencies (i.e., for which trait an agent received a positive/negative payoff) were inherited or maybe even completely fixed throughout generations. This could be maybe more clearly communicated.

Payoff-contingencies (for clarity of exposition, we now call them 'payoff profiles') are not inherited but rather drawn anew for each individual. We realised that although we explained this in the Methods section, this information was missing from the 'The Model' section - we have added clarifications there and refer to the Methods for details (lines 155-156, and 236-237).

(minor)

Page 14: "we increased the variance of this ..." (small typo)

We fixed this; we thank the reviewer for their careful reading.

Page 3: The authors often refer to the two parameters as "genes" here. From my understanding, the authors do not literally believe that the S and delta parameter are literally encoded as genes in humans. Maybe it would make sense to more clearly communicate on

this page that “genes” is more like a metaphor here for two independent strategy parameters and should not be taken to literally.

We thank the reviewer for this observation - this may indeed not be immediately obvious to all readers. We have added a clarifying sentence in the The Model section (lines 89-93).

Reviewer #3:

This paper is almost impossible to read. The terms used in the paper: “efficacy of social learning”, “flexibility”, “polarization”, “assortment”, “reliance on social learning”, total of 200 replicates (line 451), 20 randomly chosen applicates (line 171), 1,000 individuals (line 67), 20 randomly chosen individuals (line 173).

We thank the reviewer for their candid criticism of our paper. It has helped us realise that there was quite some scope for improving readability (although it was already quite readable for some audiences, see comments of Reviewer 1). The fact that the reviewer took the time to go through our manuscript even though they found it difficult to parse has really helped us improve it - many thanks for this. In response to this reviewer’s comments, we have critically evaluated (and updated) our terminology, included clear definitions where we introduce terms, and rewritten complete sections of the text to improve readability (details below). We have also made some other changes (clear overview of parameters, extensive robustness tests) that make clearer that our model choices were concisely focused on exposing a specific evolutionary principle, and that the conclusions we draw are not very sensitive to the specific choices we have made.

Regarding the terms that the reviewer mentions above, we have taken a dual approach. First, we have replaced some terms in favour of new terms that are more intuitively understandable. Second, we have taken more care to carefully explain our terms at the point in the text where we introduce them. Specifically, our revision includes the following changes:

- Throughout, the term ‘efficacy’ has been changed to ‘effectiveness’. We think this is a somewhat more intuitive term that will make our paper easier to understand for a broad readership. Accordingly, in Fig 2, we have changed ‘social learning efficacy’ to ‘effectiveness of social learning’, which is easier to parse. In addition, we have now carefully explained what we mean by this term on lines 145-152.
- We have left the term ‘flexibility’ intact - we don’t think this term can be much improved for readability without muddling too much what we exactly mean by it here. However, in our revision, we have taken care to introduce this term as clearly as possible in the context of our model (lines 71-76).
- ‘Polarization’ has been changed to ‘clustering’, and we added an extensive description of what we mean by it (lines 242-251). Upon reflection, and also based on the comments by reviewer 1, we realised that the term ‘polarization’ was not ideal, since it suggests that it refers to the population itself, while in our model it refers to how the payoffs are clustered - we thank the reviewer for flagging this.
- We have not changed the term ‘assortment’, but we have now more explicitly defined what we mean by this term. In addition, we have rewritten our remarks about this to make it more intuitive (lines 293-303).

- We have clarified our introduction of the term 'reliance on social learning' (lines 79-81).
- Instead of using the word 'replicates', we now say 'replicate simulation runs' to make more explicitly clear what we mean by this (line 522 and in the captions of Figures 2 and 4). In addition, we now explain what we mean by replicates (and how many we ran) at the end of the 'The Model' section (lines 140-141); we presume that by 'applicates', the reviewer means 'replicates'.
- Regarding '1,000 individuals', we were not entirely sure what aspect of this the reviewer considers to be problematic, but we have introduced more explicitly now that we created an individual-based model in which we explicitly simulate the evolution of a population of 1,000 individuals (lines 67-69).
- For improved readability, we have changed '20 randomly chosen individuals' to '20 individuals who were randomly chosen from the population' (lines 200-201).

Figure 2, which is very hard to read, purports to show much of the evolution of S , which appears to be the probability of learning socially. But the additional term "efficiency of social learning" is not defined in the text, but seems to turn up in Methods (line 399). Here, however, it has a separate paragraph which does not mention what "average social learning efficacy" is nor what "individual variation" in it is. It is "graphed" in Figure 2a, b, but again there is no definition. Does it relate to payoffs? Scenario 1 in Methods is totally confusing.

We thank the reviewer for pointing this out - it has helped us realise that our description of Figure 2 was not very accessible for readers that may be less acquainted with simulation studies like ours. This is a crucial part of the text where the reader is first presented with our simulation results, and so it is imperative that we help the reader to the best of our ability to parse how we arrive at the results presented here.

For this reason, we have completely rewritten this part of the Results section ('Impact of the effectiveness of social learning', lines 143-218) to improve clarity and readability. We have also rewritten the Methods section for this part 'Scenario 1: social learning effectiveness' (lines 466-477). Although this part is necessarily more technical than its counterpart in the Results section, we believe that the readability of this part has also significantly improved. We think this constitutes one of the more important improvements of the paper after review - we thank the reviewer for helping us make this paper more accessible.

Scenario 2 in Methods introduces polarization without a qualitative justification. Under what empirical situation would this be relevant? On the other hand, assortment seems to be related to what in most of the literature is called "conformity bias" or "frequency dependent bias". Does this lead to polarization?

We had indeed not provided much empirical justification for varying the degree of payoff clustering (formally called polarization, see first point of this reviewer) in our model. We have now updated the manuscript with two examples where clustering might be high and two where it might be low (lines 242-251). We believe that our revised description makes it clear that assortment is not quite the same as conformity bias or frequency dependent bias in general. While assortment describes the situation where individuals preferentially learn from individuals that they are similar to, frequency-dependent bias describes the situation where individuals adopt cultural variants based on their frequency in the population (conformity is

the case where individuals disproportionately adopt traits that are more frequent). We have now more clearly defined assortment from the outset to avoid confusion on this (lines 293-305).

The confusion over social learning efficiency is amplified in lines 187-198: “We assume all individuals have an equal social learning efficiency”. But on line 199-206 it is differences in “reliance on social learning” that matter. Also, how is “individual variation” in social reliance assessed to start with? Again, on line 228, what is “developmental flexibility”? Is it just Δ ?

We thank the reviewer for pointing out that this was confusing. Indeed, in the first scenario we vary the **effectiveness** (new term in this revision to improve readability, see response to point 1) of social learning between individuals. As described above, we have completely rewritten this part of the Results, which we believe should resolve confusion around this.

Unlike in the first scenario, the effectiveness of social learning is not varied between individuals in the second scenario. Instead, we vary the **payoffs** that they receive for adopting cultural traits. We have now stated this more explicitly where we introduce the second scenario in the Results section (lines 222-236). The variation in **reliance on social learning** is the outcome variable that we are interested in across all scenarios we consider in our simulations. We firmly believe that our rewritten version of scenario 1 of the Results section now makes this much more clear.

The reviewer is correct that developmental flexibility is Δ . As described above after the first point of this reviewer, we have now made this clearer from the outset, where the term is introduced. For extra clarity, we now make clear that we use both ‘developmental flexibility’ and ‘flexibility’ for Δ where we introduce this parameter (line 85).

In the inheritance of social learning (S and Δ are referred to as genes, which they obviously are not), it appears that generations are non-overlapping, but what are the properties of S and Δ at the outset of generation $t+1$ in terms of the parental generation t ? Is there a new value of S_0 at the beginning of each generation? Does it depend on variation in S_{10} from generation t ? How is Δ inherited if at all?

We thank the reviewer for making this important point. It is indeed true that we do not think of S_0 and Δ as being literally genes. We have now explained this right at the point where we introduce these variables (lines 89-93; see also response to Reviewer 2).

It is also true that in our simulations, the generations are non-overlapping. We now explicitly point this out in the revision (line 416). When individuals reproduce, S_0 and Δ are inherited by their offspring with a small chance of mutation (lines 135-136, and 428-435). The value of S_{10} has no bearing on the inheritance of S_0 - we have now made this explicitly clear in the Methods section (lines 435-437).

The authors do not seem to be aware of Wakano et al. *Theor. Popul. Biol.* 66: 249-258 or Aoki et al. *Curr. Anthropol.* 46: 334-340 on the evolution of social learning in varying environments.

We thank the reviewer for drawing our attention to these papers. The mentioned papers present a mathematical analysis of how the speed of environmental changes may lead to evolution of individual learning, social learning, or innate determination of behaviour. In each of these papers, individuals' learning strategies are under genetic control and cannot change over the individual's lifetime. Although these papers are relevant in that they also consider responses to uncertainty through different learning strategies, the aim of our paper is quite different.

First, our main aim is explaining the widely observed individual differences in reliance on social (as opposed to individual) learning. Second, we provide an explicit formal model for how developmental flexibility may help individuals attune their learning strategies to the specific circumstances that they face - the papers mentioned above do not consider any type of flexibility. Third and finally, the source of uncertainty in our model stems from the constitution of the population, that is, the (un)predictability driving many of the results is explicitly social, rather than environmental.

Nevertheless, we agree with the reviewer that these papers are relevant to contextualise our results because they also discuss the impact of uncertainty on adaptation. We have therefore cited these papers in the Discussion section (line 363).

The title of the paper is misleading. The assumptions, although very confusing, seem to include many about individual differences, for example, lines 366-371 and 379-389. With these assumptions that there are individual differences, how can they "emerge"?

We were puzzled by this comment. There is not much talk about individual differences in lines 366-389 of our original document. The only point in these lines that relates to individual variation is the mutation rate, which indeed introduces individual variation in the population. However, the variation introduced by mutation is essential for any evolutionary model (and the evolutionary process in general), because no change is possible without variation to select from. Also, our mutation rate is too low to generate large amounts of genotypic variation, especially in a relatively small population in which genetic drift acts to reduce variation (mutations occur with a probability of 0.001).

In our model, the individual variation with respect to reliance in social learning arises because the same developmental programmes (determined by S_0 and Δ) play out differently for different individuals, because they have different experiences with social and individual learning throughout their lives. It is true that these differences are ultimately rooted in individual differences on another level, such as differences in the effectiveness of social learning (scenario 1) or differences in the payoffs that individuals receive (scenarios 2 and 3). However, these differences do by no means **directly** cause individual differences in reliance on social learning. Rather, they provide the selective environment in which individual differences in social learning evolve through flexible developmental programs. Based on these considerations, we disagree that the title is misleading, and have left it intact.

Abstract, lines 19-20. The two papers referred to above answered the question by showing that if environments are very predictable or very unpredictable then social learning has no advantage, while in-between it does have an advantage.

We agree that these papers are relevant (and have now cited them, line 363) but they also have quite a different aim from our paper; see our response above.

The kind of simulation study where you throw in the whole “kitchen sink” of parameters and variables is very unlikely to yield useful qualitative conclusions. This paper does not advance theory of social evolution.

We do not share the view that we threw the whole ‘kitchen sink’ of parameters and variables at this study. Our model was carefully constructed to study the operation of one specific principle: when individuals have uncertainty about how well social learning will work for them, flexible developmental programs have a selective advantage, with individual differences in the reliance on social learning as the ultimate result. We have not included any aspects in our model that are unnecessary for showing this specific principle in action. However, it is true that we present three different scenarios to show this same principle in action, which the reviewer perhaps considers excessive. This is fair, but we think we have good reasons for doing this.

First, by presenting three scenarios, we could work our way up from a more simple (and therefore more easily understandable) first model to more intricate models. We consider the first model to be mostly illustrative, but also somewhat obvious, as we concede in the main text. But it does allow us to slowly build up to more interesting models where we do not directly assume that social learning is more effective for some than for others, but where this indirectly arises from the population constitution. We do admit that our original description of the first scenario left quite some scope for readability improvement, and was therefore not achieving its goal of slowly easing the reader into understanding our results in a very effective way. As laid out above, we have completely rewritten the part of the Methods and the Results section for this scenario, and believe these parts to now achieve this goal much better. We think this will help with showing that the model is not just throwing various arbitrary elements at the problem, but is in fact focused on exposing a specific principle.

Second, by having three models, we could explicitly show that the same principle (flexibility being selected because individuals are uncertain about how well social learning will work for them) can play out in different ways. This helps us point out the generality of the mechanism that we observe.

Having said all this, we concede the point that we had not very clearly outlined the parameters of our model. This might have given the impression that the model is big and unwieldy. We have now added a table with all model parameters in the Methods section (Table 1), which we think should help the reader understand that our model is not overly complicated (see also response to Reviewer 2).

Finally, we have run many additional simulations to check the robustness of our results with respect to several parameters and implementation choices (see responses to Reviewer 2). We generally find that our conclusions are highly robust, which shows that our results are quite general and not too dependent on specifics of implementation.

Reviewers' Comments:

Reviewer #2:

Remarks to the Author:

I now had a chance to read the revised manuscript and Supplementary Information and the response letter. The authors have done an excellent job addressing all the points raised during the initial submission. I particularly appreciate the calculations in the Supplementary Note, which clearly outline the underlying trade-off. As a result, I have no further comments and would like to congratulate the authors on this interesting study.

Minor:

Line 232: Parenthesis missing

Line 357: maybe consider putting genes in parentheses (i.e., "genes")

Perhaps an interesting implication of the results (line 371 following or 401 following):

If assortment is endogenous instead of fixed (as in the simulations), differences in payoffs may cause "echo chambers." In other words, if there is high heterogeneity in payoffs and agents can decide who to learn from (i.e., some form of endogenous assortment), they may become "selective social learners," learning only from others that share their "preferences"/payoff-structure. This could be worth mentioning for future work.

Reviewer #3:

Remarks to the Author:

The authors have improved the draft and removed the ambiguities. The writing is much tighter. One clarification should be made. On line 449, the probability of incorrectly concluding that a detrimental trait is beneficial or vice versa is 0.159. In the supplement the number is 0.16. This should be fixed.

Response to reviewers for the manuscript ‘Individual differences in social learning can evolve when the benefits of social information are unpredictable’

By Pieter van den Berg, TuongVan Vu, and Lucas Molleman

Below, we reprint the final comments and suggestions (after the second round of review) of the reviewers in blue. Our responses are printed in black.

Reviewer #1

(no comments)

Reviewer #2

I now had a chance to read the revised manuscript and Supplementary Information and the response letter. The authors have done an excellent job addressing all the points raised during the initial submission. I particularly appreciate the calculations in the Supplementary Note, which clearly outline the underlying trade-off. As a result, I have no further comments and would like to congratulate the authors on this interesting study.

We thank the reviewer for their kind words and their constructive comments on our manuscript.

Minor:

Line 232: Parenthesis missing

Fixed.

Line 357: maybe consider putting genes in parentheses (i.e., “genes”)

As we understand it, this is in conflict with *Nature Communications* editorial policy, so we have refrained from making this change.

Perhaps an interesting implication of the results (line 371 following or 401 following):

If assortment is endogenous instead of fixed (as in the simulations), differences in payoffs may cause "echo chambers." In other words, if there is high heterogeneity in payoffs and agents can decide who to learn from (i.e., some form of endogenous assortment), they may become "selective social learners," learning only from others that share their "preferences"/payoff-structure. This could be worth mentioning for future work.

This is an interesting perspective – we suspect that the possibility for the evolution of endogenous assortment might lead to relatively high assortment if there is high heterogeneity in payoffs, thereby potentially reducing the scope for developmental flexibility (depending on the respective costs of flexibility and endogenous assortment). We have taken the suggestion of the reviewer and have mentioned this for future work in lines 386-388.

Reviewer #3

The authors have improved the draft and removed the ambiguities. The writing is much tighter. One clarification should be made. On line 449, the probability of incorrectly concluding that a detrimental trait is beneficial or vice versa is 0.159. In the supplement the number is 0.16. This should be fixed.

We thank the reviewer for their contribution to this manuscript. We have updated the number in the Supplementary Information to 0.159 for consistency with the main text.